# Impact of Indocyanine Green Dose on Sentinel Lymph Node Mapping in Cervical Cancer: A Systematic Review

**DOI:** 10.3390/cancers16173107

**Published:** 2024-09-08

**Authors:** Joel Laufer, Santiago Scasso, Andrea Papadia

**Affiliations:** 1Department of Gynaecology, Gynecologic Oncology Unit, Hospital Británico, Montevideo 11600, Uruguay; sscasso@gmail.com; 2Department of Gynaecology, Gynecologic Oncology Unit, Hospital Casmu, Montevideo 11600, Uruguay; 3Department of Gynaecology and Obstetrics, Ente Ospedaliero Cantonale, 6500 Lugano, Switzerland; andrea.papadia@eoc.ch; 4Facoltà di Scienze Biomediche, Università della Svizzera Italiana, 6900 Lugano, Switzerland

**Keywords:** cervical cancer, sentinel lymph node, indocyanine green

## Abstract

**Simple Summary:**

Sentinel lymph node (SLN) mapping is a crucial technique in the surgical treatment of cervical cancer, helping to identify the first lymph nodes that cancer may spread to. Indocyanine green (ICG) is a dye commonly used to make these lymph nodes visible during surgery, but there is no agreement on the best amount to use. This research reviews different amounts of ICG used in previous studies to determine the most effective dose for detecting these lymph nodes. By identifying the best practices, this study aims to improve the accuracy and safety of surgeries for cervical cancer patients. The findings could help standardize the use of ICG in clinical practice, leading to better outcomes for patients and providing clearer guidelines for surgeons.

**Abstract:**

Over the past decade, SLN mapping has become increasingly important in cervical cancer surgery. ICG is the most commonly used tracer due to its high bilateral detection rates, ease of use, and safety. However, there is no consensus on the optimal ICG dose, leading to variability in outcomes. This systematic review aims to evaluate the impact of different ICG doses on SLN detection in early-stage cervical cancer, identifying the most effective and safe dose for clinical practice. A comprehensive search was conducted in MEDLINE/PubMed up to May 2024. Studies included assessed SLN mapping using ICG in stage IA2-IIA/IIB cervical cancer. Exclusions were applied to studies not reporting ICG dose or using multiple tracers without dose-specific results. Twelve studies were included, with ICG concentrations ranging from 0.25 mg/mL to 25 mg/mL and injection volumes from 1 to 10 mL. Overall SLN detection rates ranged from 88% to 100%, while bilateral detection rates varied between 74.1% and 98.5%. The most consistent results were obtained with an ICG concentration of 1.25 mg/mL and a 4 mL injection volume. In conclusion, an ICG concentration of 1.25 mg/mL with a 4 mL injection volume is recommended for effective SLN mapping in cervical cancer, achieving high detection rates with minimal variability. Standardizing this dose in clinical practice is suggested to improve reproducibility and outcomes.

## 1. Introduction

Cervical cancer is one of the most common female cancers worldwide, with approximately 604,000 new cases diagnosed annually [1]. The distribution of this cancer varies significantly across different regions. In developed countries, where HPV vaccination and screening are widely implemented, the incidence and mortality rates of cervical cancer are very low [2]. In these settings, cervical cancer is often diagnosed at an early stage, making surgical treatment a viable option [2].

The main surgical treatment involves the removal of the uterus through either simple or radical hysterectomy, depending on the cancer’s size and pathological features, and the removal of lymph nodes [3]. Lymph node status is a crucial prognostic factor for survival in cervical cancer, highlighting the importance of nodal assessment. In cases of lymph node metastasis, adjuvant treatment is indicated [3].

Over the past two decades, sentinel lymph node (SLN) mapping has become increasingly common in cervical cancer management. This technique offers targeted sampling of the key lymph nodes draining the uterus, providing an excellent alternative to full lymphadenectomy [4]. The precision of SLN mapping, combined with the pathological ultrastaging, which is possible only with SLNs and not a large number of lymph nodes, makes it a reliable surgical staging tool that is progressively replacing lymphadenectomy, thereby reducing surgical morbidity [5].

Pathological ultrastaging of SLNs is essential, as it identifies N1 (macro and micrometastasis) involvement in approximately 43% of cases [5]. Failure to detect lymph node involvement could result in omitting adjuvant radiotherapy in otherwise low- or intermediate-risk tumors, which increases the risk of disease recurrence. Additionally, frozen section examination can assess lymph node status prior to radical uterine surgery, enabling the substitution of radical surgery with primary chemoradiation in the presence of lymphatic metastases, thus avoiding the morbidity associated with dual-modality treatment [3].

From an oncological point of view, a crucial question is whether complete lymph node dissection is necessary, or a SLN biopsy alone suffices, as assessed by disease-free and overall survival rates. The SENTIX trial, designed as a prospective, multicenter cohort study, sought to determine if a SLN biopsy is comparable to systematic pelvic lymphadenectomy in preventing recurrence in early-stage cervical cancer [6]. All patients underwent a bilateral SLN biopsy with frozen section analysis, followed by either hysterectomy or trachelectomy. Those with SLN-positive results on frozen section were excluded [6]. Cibula recently shared the final results at the 25th European Congress on Gynaecological Oncology: disease-free survival at 2 years was 93.3%, and overall survival was 97.9%. During a median follow-up of 47 months, 54 patients (9.1%) experienced recurrence, and 20 (3.4%) died [7]. The study revealed that a SLN biopsy provided 2-year disease-free and overall survival rates comparable to those seen with pelvic lymphadenectomy [7]. The authors concluded that a SLN biopsy with ultrastaging, without further lymphadenectomy, did not increase the risk of recurrence. Furthermore, analysis from two multicenter trials (SENTICOL I and II) confirmed that skipping systematic pelvic lymphadenectomy in cases with bilateral negative SLNs did not increase recurrence risk [8]. The outcomes of the ongoing SENTICOL III study, whose primary objective is to compare 3-year disease-free survival and health-related quality of life following a SLN biopsy versus a SLN biopsy combined with pelvic lymphadenectomy in early cervical cancer, are highly anticipated [9].

For SLN procedures to be reliable, they must achieve a high bilateral detection rate (DR) (at least one SLN detected in each hemipelvis) and a low false-negative rate (FNR) (tumor-negative SLNs concurrent with tumor-positive non-SLNs at lymphadenectomy), thus minimizing the risk of undertreating patients [10].

Currently, two primary methods are employed for SLN detection in cervical cancer: the conventional combination of the radiotracer technetium-99m nanocolloid (99mTc) and blue dye, and the more recent use of indocyanine green (ICG) [11]. The 99mTc enables preoperative imaging with SPECT-CT and intraoperative detection of radioactive signals, while blue dye assists in visualizing lymphatic architecture and SLNs during surgery. ICG, visualized using near-infrared (NIR) fluorescence imaging, provides real-time navigation with better tissue penetration compared to blue dye. Its feasibility and high detection rates in early-stage cervical cancer have been demonstrated, and its favorable safety profile and logistical advantages make it a preferable option over the 99mTc and blue dye combination [11].

The adoption of ICG as tracer has significantly advanced SLN mapping. Its ease of use and intrinsic fluorescent properties have facilitated its incorporation into clinical practice for both cervical and endometrial cancer [12]. International guidelines [3,13] and Delphi expert opinions [14] recommend using ICG for SLN mapping in cervical cancer. The expert consensus has established recommended surgical steps for SLN dissection in cervical cancer, potentially serving as a quality assurance tool to standardize surgical practice and clinical trials [14]. The consensus identified 15 recommended, three optional, and five not recommended steps, with 100% agreement on using ICG as the radiotracer of choice. However, there was no agreement on the ICG concentration and the total volume of tracer injected into the cervix, as these have been primarily investigated in endometrial cancer [14]. Therefore, the optimal dose of ICG for effective SLN mapping in early-stage cervical cancer remains undefined. Various institutions have adopted different dosages for SLN mapping in cervical cancer. In this systematic review, we evaluate the impact of different dosages of ICG on SLN mapping in patients with early-stage cervical cancer.

## 2. Methods

This systematic review adhered to the guidelines set forth by the Preferred Reporting Items for Systematic Reviews and Meta-Analyses for Diagnostic Test Accuracy (PRISMA-DTA) statement, which provides an evidence-based checklist for reporting in systematic reviews and meta-analyses of diagnostic studies [15].

### 2.1. Search Strategy

Two authors (JL and SS) performed a comprehensive computer-based literature search of the National Library of Medicine’s MEDLINE/PubMed database for articles published from inception to May 2024. No beginning-date limit restrictions were used.

### 2.2. Study Selection

The following keywords were used: “cervical cancer”, “sentinel lymph node”, and “indocyanine green”. Articles relevant to the subject in the citations of each report were additionally included. Studies were excluded if they were case reports, conference abstracts, reviews, or short communications, because they do not provide sufficient information to assess the methodological quality. Uncertainties were resolved in consensus. Studies were considered if they were original reports on the outcome of ICG SLN mapping in patients with early-stage cervical cancer undergoing surgery with curative intent (stage IA2-IIA and IIB).

Studies were also excluded if they met any of the following criteria:SLN mapping was performed after neoadjuvant chemotherapy or radiotherapy.Endometrial and cervical cancer results were combined and could not be separated.Multiple tracers were used and results for each single tracer were not available.The dose of ICG was not reported.The dose was changed throughout the study and results between doses could not be separated.

### 2.3. Data Extraction

For each eligible article, information was collected concerning the basic study details (authors, year of publication, study design), patient characteristics (number of patients evaluated, mean/median age, body mass index (BMI), stage according to the International Federation of Gynecology and Obstetrics (FIGO), tumor size), type of surgical approach (laparoscopy, robotic, laparotomy), technical aspects (ICG concentration mg/mL and ICG volume mL) and outcomes (number of SLNs, overall DR, bilateral DR).

### 2.4. Quality Assessment

The quality of the studies included in this review was thoroughly evaluated using the updated “Quality Assessment of Diagnostic Accuracy Studies” (QUADAS-2) tool [16]. This instrument examines four key areas: patient selection, index test, reference standard, and the flow and timing of the study. Each of these domains was carefully assessed for potential bias, and the findings were visually represented in a summary graph [16]. A detailed overview of the quality assessment is provided in Figure 1.

## 3. Results

### 3.1. Literature Search

The initial search yielded 187 articles, all in English. From these, 147 records were excluded initially based on title and abstract. According to the inclusion and exclusion criteria, the following records were removed from the remaining 40 articles: 5 duplicates, 12 articles combining endometrial and cervical cancer, 10 articles using multiple tracers, and 1 article for FIGO stage greater than IIB. After the removal of these records, 12 studies were eligible and included in the systematic review (Figure 2: Flow chart of selected studies).

### 3.2. Basic Study and Patient Characteristics

Twelve articles evaluating the use of ICG and SLN in cervical cancer were selected [17,18,19,20,21,22,23,24,25,26,27,28]. The selected articles were published between 2015 and 2024 by researchers from Europe (*n* = 8), the US (*n* = 2), Japan (*n* = 1), and the Republic of Korea (*n* = 1). Of the 12 included articles, 10 were retrospective cohort studies (2 multicenter) and 2 were prospective cohort studies (1 multicenter). The number of patients included ranged between 20 and 245; the mean/median age of patients ranged between 39 and 51 years; the median BMI ranged between 22.4 and 28.8. Most of the patients included had a FIGO stage of IB1 tumor. Regarding tumor size, the majority of the studies (9 out of 12) classified the dimension of the tumor as either less than or greater than 2 cm. Of these 9 studies, 51.7% of the patients had tumors smaller than 2 cm, and 48.3% had tumors larger than 2 cm. Of the remaining three studies, one did not have data regarding tumor size, another classified the diameter as greater or less than 4 cm, and the last one only reported a median tumor size of 14 mm. The majority (87%) of the SLN procedures were performed using a minimally invasive approach (47% robotic and 40% laparoscopy), and only 13% of the procedures were performed via laparotomy (Table 1: Basic study and patient characteristics).

### 3.3. Methodological and Technical Aspects of the Included Studies

The twelve studies included in this systematic review exhibit a variety of methodological and technical approaches to the detection of SLN using ICG in cervical cancer patients.

#### 3.3.1. Study Design

The studies encompassed retrospective cohort studies (*n* = 10, including 2 multicenter) and prospective cohort studies (*n* = 2, including 1 multicenter) (Table 1). The retrospective studies reviewed past patient records to analyze outcomes, while the prospective studies followed patients forward in time to gather data.

#### 3.3.2. Inclusion and Exclusion Criteria

All studies selected patients based on specific inclusion and exclusion criteria. Common inclusion criteria were patients with cervical cancer stages IA1 to IB1, as per FIGO classification, and candidates for SLN mapping. Exclusion criteria often included patients with prior pelvic radiation, presence of bulky lymph nodes detected on imaging, and those with contraindications to ICG.

#### 3.3.3. Intervention and Procedures

The administration of ICG varied across studies. Dosages ranged from 0.5 to 4 mL of ICG, typically at concentrations of 0.5 to 1.25 mg/mL. The injection sites were generally the cervix, either at 3 and 9 o’clock positions or in a circumferential manner around the tumor. The procedures for SLN mapping included robotic-assisted surgery, laparotomy, and laparoscopy, with some studies using a combination of laparoscopy and laparotomy approaches. The timing of the ICG injection relative to the surgery also varied, from immediate preoperative injection to several hours prior.

#### 3.3.4. Outcome Measures

The primary outcomes measured across studies included SLN DRs, sensitivity, specificity, and FNRs. Secondary outcomes often included the number of SLNs identified, bilateral DRs, and the complications associated with the ICG injection and a SLN biopsy.

#### 3.3.5. Statistical Analysis

Various statistical methods were employed to analyze the data. Descriptive statistics were used to summarize patient characteristics and outcomes. Comparative analyses, such as chi-square tests or t-tests, were conducted to compare detection rates and other outcomes between different subgroups. Multivariate logistic regression models were often used to adjust for potential confounders and to identify factors independently associated with successful SLN detection.

### 3.4. Main Findings

#### 3.4.1. ICG Dose

Consistent dosage of 1.25 mg/mL: The majority of studies (10 out of 12) utilized an ICG concentration of 1.25 mg/mL. This consistency suggests a standard dosage across many studies (Figure 3).

Volume variability: The volumes used in these studies varied significantly. Most studies used a volume of 4 mL [21,22,23,24,25,26]. Some studies used higher volumes of 8–10 mL [17,19,20].

Outliers in dosage and volume: Smits [27] used a significantly lower ICG concentration (0.25 mg/mL) with a volume of 1 mL. Persson [28] used a significantly higher ICG concentration (25 mg/mL) with a volume of 1 mL (Figure 4).

#### 3.4.2. SLN Detection

Table 2 shows that the median/mean number of SLNs detected per study ranges from two to six SLNs. The overall DRs of the SLNs range from 88% to 100%, with most studies reporting rates above 90%. Bilateral DRs show greater variability, ranging from 74.1% to 98.5%. FNRs vary considerably between studies, from 0% to 23.08%. The negative predictive value (NPV) is high in most studies, indicating that the probability of a negative result being correct is elevated (Table 2).

#### 3.4.3. Integrated Analysis of Results

The majority of studies use an ICG concentration of 1.25 mg/mL with varying volumes ranging from 1 mL to 10 mL. Two studies use concentrations outside this range: Smits [27] (0.25 mg/mL) and Persson [28] (2.5 mg/mL). Regarding overall DRs and ICG dosage, eight studies [17,18,19,20,21,22,25,26] report high DRs (≥95%), and four studies [14,23,27,28] report moderate to high DRs (88–94.5%).

With regards to bilateral DRs and ICG dosage, higher bilateral DRs (≥88%) were observed in studies using ICG concentrations of 1.25 mg/mL with varying volumes. Lower bilateral DRs (74.1–85.4%) were observed in some studies with the standard ICG concentration but differing volumes, suggesting that other factors, such as surgical technique, may also play a significant role (Table 3 and Table 4). Concerning the FNR and ICG dosage, studies using a standard concentration of 1.25 mg/mL generally report low FNR (0–3.7%), except for a few outliers: Kim [21] (1.25 mg/mL, 4 mL, FNR 23.08%) and Papathemelis [22] (1.25 mg/mL, 4–5 mL, FNR 16.7%). Finally, high NPVs (≥97%) were reported across 10 studies using the standard 1.25 mg/mL concentration with varying volumes, except for 2 studies: Di Martino [20] (NPV of 93.7%) and Kim [21] (NPV of 92.4%).

## 4. Discussion

### 4.1. Summary of Main Findings

Our systematic review indicates that the use of ICG at a concentration of 1.25 mg/mL, regardless of the volume, generally results in high DRs of SLNs in cervical cancer. Variability in volumes used did not significantly impact the detection efficacy.

### 4.2. Interpretation of Results

The high DRs observed across multiple studies confirm the effectiveness of ICG in SLN mapping. Given that only two studies [27,28] reported results with a different ICG concentration, it is difficult to comment that one concentration is superior to another. However, given the good results reported with the concentration that is the most adopted, it seems very reasonable to set the concentration of 1.25 mg/mL as the standard of care. Additionally, the high DRs indicate that this standard concentration is effective across different surgical techniques and patient populations, reinforcing its reliability and utility in clinical practice. A meta-analysis published by Xiong et al. [29] showed that ICG at concentrations lower than 5 mg/mL and injection volumes of 2 mL or more significantly improved diagnostic performance in a SLN biopsy. Specifically, these conditions resulted in higher detection rates and sensitivity compared to higher concentrations and smaller volumes [29]. These findings suggest that lower concentrations and larger volumes of ICG can optimize the efficacy of detecting lymph node metastases. In this context, the FDA approved ICG for fluorescence imaging of lymph nodes and lymphatic vessels during lymphatic mapping in gynecologic cancers, including cervical and uterine tumors [30]. The recommended dose for this application is four 1.25 mg/mL injections (four 1 mL injections) for a total dose of 5 mg (4 mL) [30]. The results of SLN mapping with ICG are consistently very good across different papers. Differences in concentrations and volumes of ICG used do not seem to have a significant impact on SLN mapping performance. These results highlight the overall excellent performance of ICG SLN mapping. Its strength is probably related to the fact that it provides visual information (as opposed to audiometric information with Tc99), which is enhanced by the natural fluorescence of ICG. Notably, studies such as those by Imboden [17] and Beavis [18] reported 100% DRs using 1.25 mg/mL with varying volumes. The observed high DRs underscore the robustness of ICG at the specified concentration, allowing for flexibility in its application across different clinical settings. This adaptability is particularly valuable in accommodating diverse surgical techniques and patient anatomies, thereby enhancing the generalizability of the findings.

### 4.3. Clinical Implications

The majority of the studies have adopted a standard concentration of 1.25 mg/mL. This is probably related to the results of the meta-analysis published by Xiong [29] and the recent FDA recommendation for SLN mapping [30]. Consequently, this concentration could be considered the standard of care. The flexibility in volume administration, as evidenced by the variability in successful volumes used, allows for tailored approaches based on individual patient needs and surgical conditions. Standardizing the ICG concentration at 1.25 mg/mL across clinical settings can streamline protocols and enhance the reproducibility of results. This standardization can also facilitate training and implementation in diverse healthcare environments, ensuring that more patients benefit from accurate and effective SLN mapping. Additionally, the ability to adjust the volume based on specific surgical scenarios without compromising DRs provides surgeons with the flexibility to optimize their procedures, potentially reducing complications and enhancing overall patient satisfaction.

### 4.4. Comparison with Existing Literature

Our findings align with existing literature that supports the use of 1.25 mg/mL ICG for high DRs in SLN mapping [12]. However, the variability in bilateral DRs and FNRs indicates that factors beyond ICG dosage, such as surgical technique, patient characteristics, and tumor diameter, play a significant role. Despite the fact that the procedure to perform a SLN biopsy has been described and implemented, variations in surgical techniques have been reported and might represent an obstacle when comparing results from different studies. Standardization of technique and quality assessment tools are critical. Recently, a consensus among experts on the surgical technique to perform SLN dissection in cervical cancer was established.

Experts have established a set of recommended practices [14]: using ICG as the tracer; performing superficial injections (with or without deep injections) at the 3 and 9 o’clock positions; injecting at the margins of unaffected mucosa while avoiding the vaginal fornices; grasping the cervix with forceps only in tumor-free areas; employing a minimally invasive approach for a SLN biopsy during simple trachelectomy or conization; identifying the ureter, obliterated umbilical artery, and external iliac vessels prior to SLN removal; initiating dissection at the uterine artery level and moving laterally; and completing dissection in one hemipelvis before addressing the opposite side. Furthermore, consensus discouraged injections at the 6 and 12 o’clock positions, direct injections into the tumor when it fully occupies the cervix, removing nodes through ports without protective measures, omitting an ultrastaging protocol, and altering tracer concentration during re-injection after a mapping failure [14].

Experts did not agree on 42% of the questions, highlighting the difficulties in standardizing this surgical practice [14]. Regarding the items where there was no agreement, such as ICG concentration and total volume of tracer injected, these had been previously investigated only in endometrial cancer [31]. While ICG concentration has not been found to influence bilateral SLN mapping rates, the meta-analysis published by Xiong showed that an ICG volume of at least 3 mL was associated with a reduced mapping failure rate [29]. Concerning the timing of injection (before or after establishing a pneumoperitoneum), there was no consensus, but it must be highlighted that injecting before abdominal entry may cause ICG migration to non-SLNs, making it more difficult to identify the “true” SLN. Of the 12 studies included in this review, only 2 mention the time between injection and SLN dissection. In the first study, Bizzarri [23] mentions that about 15 min after cervical injection, the pelvic retroperitoneal space was opened, and lymph nodes were assessed with a NIR camera. In the second study, Aoki [25] describes a mean time of 24 min from ICG injection to the start of laparoscopic SLN mapping, including the creation of a vaginal cuff. The remaining 10 studies do not provide a precise time and generally mention that the ICG injection was performed after anesthetic induction. These aspects reveal that the quality of SLN technique still varied considerably in many aspects of the procedure, including the doses of ICG (volume and mL).

### 4.5. Strengths and Limitations

A major strength of our review is the comprehensive inclusion of diverse studies, providing a robust evidence base for the effectiveness of ICG in SLN mapping. The broad range of studies included allows for a thorough examination of different protocols and patient populations, thereby enhancing the validity of our conclusions. The consistency in reporting high detection rates across most studies reinforces the reliability of ICG at a concentration of 1.25 mg/mL for SLN mapping in cervical cancer. However, the heterogeneity in surgical techniques, ICG volumes, and patient populations introduces variability that may affect the generalizability of our findings. Differences in surgical expertise, techniques for ICG administration, and patient characteristics such as BMI and tumor size could lead to discrepancies in outcomes. This variability highlights the need for standardized protocols to minimize these differences and ensure more consistent results. Additionally, one study did not specify the concentration or volume used, which could impact the consistency of the reported outcomes. The lack of detailed reporting in some studies limits the ability to fully compare and synthesize the results, potentially introducing bias. Future research should aim to standardize the reporting of key variables such as ICG concentration, volume, and timing of injection to facilitate more accurate comparisons and meta-analyses. Despite these limitations, our review underscores the efficacy of ICG in SLN mapping. The findings suggest that with standardized protocols, ICG can be a highly effective tool for SLN detection in cervical cancer, leading to better clinical outcomes and patient care.

### 4.6. Recommendations for Future Research

Future research should focus on standardizing surgical techniques to minimize variability in SLN detection rates. Comparative studies that directly assess different volumes of ICG using the same concentration could provide more precise guidelines for clinical practice. Additionally, exploring the impact of patient demographics and tumor characteristics on SLN detection success rates could further refine the use of ICG in diverse patient populations. Moreover, investigations into the optimal timing of ICG injection relative to surgical procedures could help establish best practices for SLN mapping. Studies examining long-term outcomes of patients who undergo SLN mapping with ICG, including recurrence rates and survival outcomes, would provide valuable insights into the overall effectiveness and benefits of this approach. Collaborative efforts to develop and implement standardized protocols across multiple centers could enhance the generalizability of findings and contribute to a more uniform practice worldwide.

## 5. Conclusions

In conclusion, the use of ICG at a concentration of 1.25 mg/mL is effective for SLN mapping in cervical cancer, with high DRs observed across various volumes. Standardizing this concentration while allowing flexibility in volume can enhance clinical outcomes and patient care. Our findings support the integration of ICG into routine clinical practice for SLN mapping, while highlighting the need for standardized surgical protocols to ensure consistent and reliable results. Additionally, the standardization of ICG concentration can streamline training and implementation in diverse healthcare settings, promoting more uniform practices and improving the overall quality of care. Future research should continue to refine these protocols and explore factors that may influence detection rates, ensuring that SLN mapping with ICG remains a reliable and effective tool in the management of cervical cancer.

## Figures and Tables

**Figure 1 cancers-16-03107-f001:**
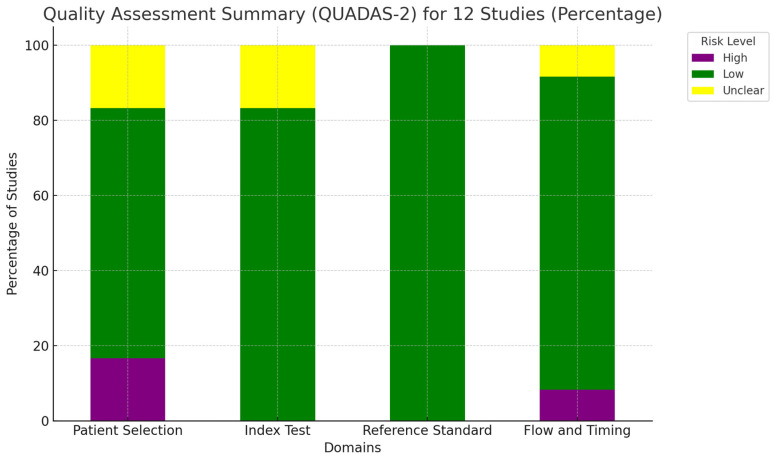
Overall quality assessment of the studies included in the systematic review, according to the QUADAS-2 tool.

**Figure 2 cancers-16-03107-f002:**
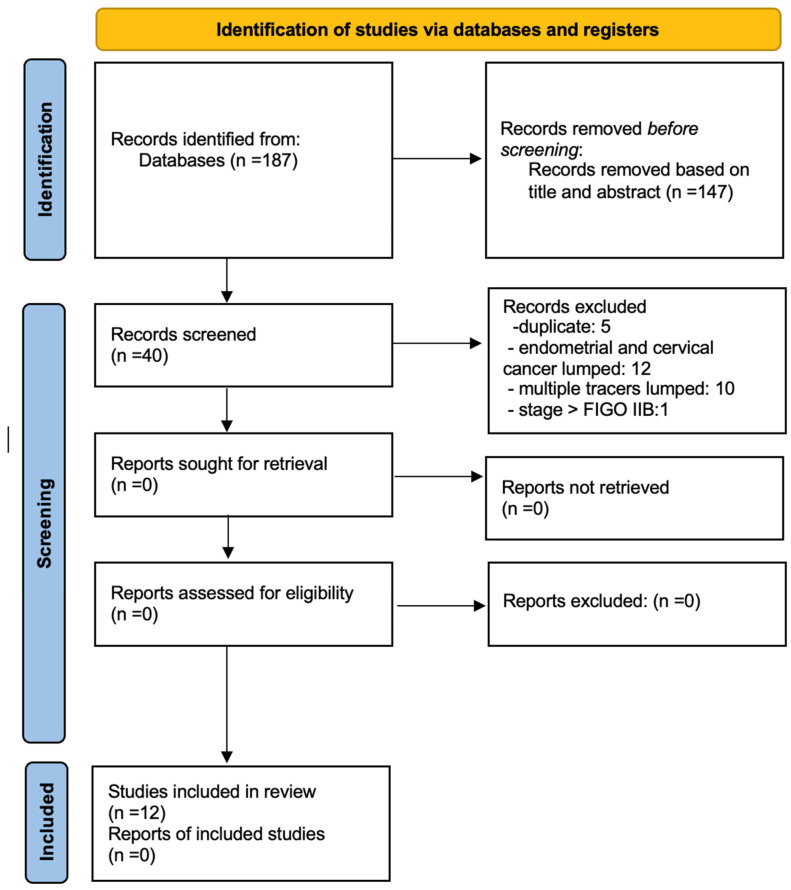
PRISMA Flow chart.

**Figure 3 cancers-16-03107-f003:**
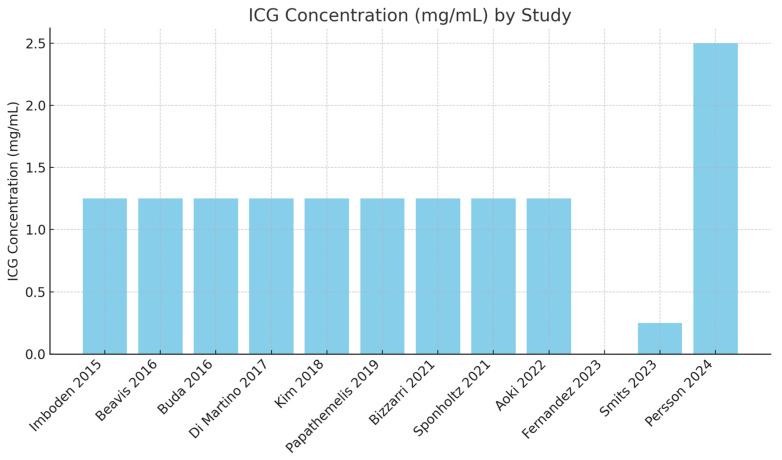
ICG concentration used in different studies [17,18,19,20,21,22,23,24,25,26,27,28].

**Figure 4 cancers-16-03107-f004:**
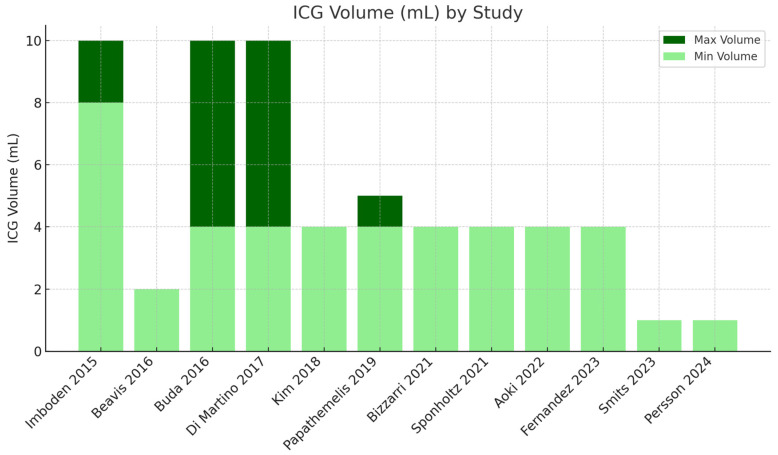
ICG volume used in different studies [17,18,19,20,21,22,23,24,25,26,27,28].

**Table 1 cancers-16-03107-t001:** Basic study and patient characteristics.

Author and Year	Type of Study	N	FIGO * Stage	Size of the Tumor <2 cm: >2 cm (%)	Age (Mean/ Median)	BMI ** kg/m^2^ (Median)	Surgical Approach (Laparoscopy, Robotic, Laparotomy)
Imboden 2015 [17]	Retrospective	22 (ICG group)	IA1 (LVSI)–IIB	<2 cm (27%) >2 cm (73%)	43	N/A	Laparoscopy (100%)
Beavis 2016 [18]	Retrospective	30	IA1 (LVSI)–IB2	<2 cm (43%) >2 cm (57%)	42	26.5	Robotic (100%)
Buda 2016 [19]	Multicenter, retrospective	68 (ICG group)	1A2–1B1	<2 cm (56%) >2 cm (44%)	42	23.7	Laparoscopy (100%)
Di Martino 2017 [20]	Multicenter, retrospective	48	IB1–IIB	>2 cm (100%)	46	24.0	Laparoscopy (100%)
Kim 2018 [21]	Retrospective	103	IA1 (LVSI)–IIA	<4 cm (76%) >4 cm (24%)	45	22.4	Laparoscopy (56%) Robotic (44%)
Papathemelis 2019 [22]	Retrospective	20	IA–IIB	N/A	51	26.0	Laparoscopy (100%)
Bizzarri 2021 [23]	Retrospective	85	IA1 (LVSI)–IIB	<2 cm (53%) >20 cm (47%)	44	23.0	Laparotomy (32%) Laparoscopy (51%) Robotic (17%)
Sponholtz 2021 [24]	Multicenter prospective cohort	245	IA2 (LVSI)–IIA1	<2 cm (53%) >2 cm (47%)	44	25.0	Robotic (100%)
Aoki 2022 [25]	Retrospective	77	IA2–IB1	<2 cm (50.6%) >2 cm (49.4%)	40	22.9	Laparoscopy (100%)
Fernandez 2023 [26]	Retrospective	106	IA1 (LVSI)–IIB	<2 cm (59%) >2 cm (41%)	40	28.8	Laparotomy (100%)
Smits 2023 [27]	Retrospective	100	IA1 (LVSI)–IIA1	<2 cm (73%) >2 cm (27%)	39	Underweight (2%) Normal (34%) Overweight (31%) Obese (28%) Unknown (5%)	Laparotomy (5%) Laparoscopy (95%)
Persson 2024 [28]	Prospective	181	IA2-IIA1	Median tumor size 14 mm	44	24.7	Robotic (100%)

* Stage according to the International Federation of Gynecology and Obstetrics. ** Body mass index. ICG—Indocyanine green. LVSI—Lymphovascular space invasion. N/A—No available.

**Table 2 cancers-16-03107-t002:** Outcomes.

Author and Year	ICG Concentration (mg/mL)	ICG Volume (mL)	Number of SLNs Median/Mean	Overall DR	Bilateral DR	FN Rate	NPV
Imboden 2015 [17]	1.25	8–10	3.7	95.5%	95.5%	0%	100%
Beavis 2016 [18]	1.25	2	2 per hemipelvis	100%	86.7%	1 enlarged non-SLN	N/A
Buda 2016 * [19]	1.25	4–5 (Monza) 8–10 (Bern)	3	100%	98.5%	0.04%	97%
Di Martino 2017 * [20]	1.25	4–5 (Monza) 8–10 (Bern)	3	100%	91.7%	11.5%	93.7%
Kim 2018 [21]	1.25	4	2.34	100%	85.4%	23.08%	92.4%
Papathemelis 2019 [22]	1.25	4–5	N/A	90%	83.3%	16.7%	N/A
Bizzarri-2021 [23]	1.25	4	2	92.6%	74.1%	N/A	96.7%
Sponholtz 2021 [24]	1.25	4	4	96.3%	82.%	3.7%	98.7%
Aoki 2022 [25]	1.25	4	2	98.7%	88.3	0%	100%
Fernandez 2023 [26]	N/A	4	4	98%	89%	0%	N/A
Smits 2023 [27]	0.25	1	3	88%	75%	2.7%	97.1%
Persson 2024 [28]	2.5	1	6	N/A	94.5%	N/A	100%

* Multicenter trials with different doses at different institutions. N/A: not available.

**Table 3 cancers-16-03107-t003:** Higher bilateral DRs (≥88%).

Author and Year	ICG Concentration (mg/mL)	ICG Volume (mL)	Bilateral DR (%)
Imboden 2015 [17]	1.25	8–10	95.5
Buda 2016 [19]	1.25	4–5, 8–10	98.5
Di Martino 2017 [20]	1.25	4–5, 8–10	91.7
Aoki 2022 [25]	1.25	4	88.3
Fernandez 2023 [26]	N/A	4	89
Persson 2024 [28]	2.5	1	94.5

N/A: not available.

**Table 4 cancers-16-03107-t004:** Lower bilateral DRs (74.1–85.4%).

Author and Year	ICG Concentration (mg/mL)	ICG Volume (mL)	Bilateral DR (%)
Beavis 2016 [18]	1.25	2	86.7
Kim 2018 [21]	1.25	4	85.4
Papathemelis 2019 [22]	1.25	4–5	83.3
Bizzarri 2021 [23]	1.25	4	74.1
Sponholtz 2021 [24]	1.25	4	82
Smits 2023 [27]	0.25	1	75

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
