# Peer review of "Impact of Indocyanine Green Dose on Sentinel Lymph Node Mapping in Cervical Cancer: A Systematic Review"

_cancers, 2024, doi:10.3390/cancers16173107_

Round 1

Reviewer 1 Report

Comments and Suggestions for Authors

Joel Laufer and coworkers reported Impact of indocyanine green dose on sentinel lymph node mapping in cervical cancer. The experimental is carefully conducted, and the results have been presented correctly, and the contents fall well into the scope of the journal. However, some points need to be addressed/answered before further considered:

1. Fig. 3: (mg/ml) should be changed to (mg/mL) . Wrong in many places.

Author Response

Fig. 3: “(mg/ml)” should be changed to “(mg/mL)” . Wrong in many places.  

Thank you for pointing this out        

We have made the corresponding changes in the figures and throughout the text.                             

Reviewer 2 Report

Comments and Suggestions for Authors

A manuscript entitled “Impact of indocyanine green dose on sentinel lymph node mapping in cervical cancer” by Joel Laufer et al., as a systematic review article was submitted to cancers for possible consideration of publications. In this study, the authors reviewed the different doses (concentrations and volumes) used for cervical cancer SLN mapping and compare the 12 results in terms of overall and bilateral detection rates, number of retrieved SLNs, false negative rate and negative predictive value. Herein, there were some major concerns to be carefully addressed before further process of the manuscript.

1)      The authors in this study, reviewed different amounts of ICG used in previous 12 studies of cervical cancer, to determine the most effective dose for detecting these lymph nodes. By summarizing the listed practices, this study aimed to improve the accuracy and safety of surgeries for cervical cancer patients. I believed that the findings could help standardize the use of ICG in clinical practice of cervical cancer only, because other cancers were not reviewed, involved or mentioned in this study. So please do not include the clinical diagnosis or surgeons of other cancers.

2)      Authors did very limited literature search therefore conveyed the risk to this so-called systematic review. I am afraid that I cannot agree with the point. As authors mentioned in the main texts, “The initial search yielded 187 articles, all in English. From these, 147 records were excluded initially based on title and abstract. According to the inclusion and exclusion criteria, the following records were removed from the remaining 40 articles: 5 duplicates, 12 articles combining endometrial and cervical cancer, 10 articles using multiple tracers, and 1 article for FIGO stage greater than IIB. After the removal of these records, 12 studies were eligible and included in the systematic review.” Only 12 literatures were used for this systematic review, I have to say that numbers are very limited.

3)      In terms of the ICG doses, in Figure 3, the main dose is 1.25 mg/ml, but the ICG dose of Fernandez 2023 was none, and the ICG dose of “Smits 2023” was 0.25 mg/ml, and dose of “Persson 2024” was even up to 2.5 mg/ml. The comparison of ICG doses should be conducted to give a clue of reasons why they used lower dose or higher dose in the clinical as vital technical approach to detect SLN in cervical cancer patients. Conditionally or randomly? Why not use mg/ml instead of mg/Kg?

4)      Regarding the ICG volume used in the studies, why a bigger volume (10 ml) or a less volume (2 ml) were used in those studies, should be described from the practical views. And why Smits 2023 and Persson 2024, both used only 1ml ICG? sufficient or not enough?

5)      No image data were cited and compared in this systematic review, why?

6)      The statistical analysis was simple. No graphical abstract.

I may here recommend this manuscript to be substantially revised.

Author Response

Thank you for your useful comments and suggestions of our manuscript

  1. The authors in this study, reviewed different amounts of ICG used in previous 12 studies of cervical cancer, to determine the most effective dose for detecting these lymph nodes. By summarizing the listed practices, this study aimed to improve the accuracy and safety of surgeries for cervical cancer patients. I believed that the findings could help standardize the use of ICG in clinical practice of cervical cancer only, because other cancers were not reviewed, involved or mentioned in this study. So please do not include the clinical diagnosis or surgeons of other cancers.

We agree with your observation that the focus of our study is specifically on cervical cancer, and our intention was to provide insights that could help standardize the use of ICG in this context alone. We have selected only the studies that included cervical cancer for the analysis. However, we have cited some articles that include both endometrial and cervical cancer, as they are very important in validating the use of indocyanine green.

2. Authors did very limited literature search therefore conveyed the risk to this so-called systematic review. I am afraid that I cannot agree with the point. As authors mentioned in the main texts, “The initial search yielded 187 articles, all in English. From these, 147 records were excluded initially based on title and abstract. According to the inclusion and exclusion criteria, the following records were removed from the remaining 40 articles: 5 duplicates, 12 articles combining endometrial and cervical cancer, 10 articles using multiple tracers, and 1 article for FIGO stage greater than IIB. After the removal of these records, 12 studies were eligible and included in the systematic review.” Only 12 literatures were used for this systematic review, I have to say that numbers are very limited.

Thank you for pointing this out.

We only included articles according to the inclusion and exclusion criteria, using the following keywords: “cervical cancer,” “sentinel lymph node,” and “indocyanine green.”

3. In terms of the ICG doses, in Figure 3, the main dose is 1.25 mg/ml, but the ICG dose of Fernandez 2023 was none, and the ICG dose of “Smits 2023” was 0.25 mg/ml, and dose of “Persson 2024” was even up to 2.5 mg/ml. The comparison of ICG doses should be conducted to give a clue of reasons why they used lower dose or higher dose in the clinical as vital technical approach to detect SLN in cervical cancer patients. Conditionally or randomly? Why not use mg/ml instead of mg/Kg?

Thank you very much for your comments.The intention of this graph is to illustrate the different doses used by various authors.  This is not yet well-defined, which is the reason for this review. The doses and volumes are primarily based on the study by Xiong, which we mentioned in the discussion. Regarding the article by Fernandez, that specific data is not mentioned, which is why it is not illustrated. As for why we used mg/mL instead of mg/Kg, it is because the dilution is defined this way in all the articles.

4. Regarding the ICG volume used in the studies, why a bigger volume (10 ml) or a less volume (2 ml) were used in those studies, should be described from the practical views. And why Smits 2023 and Persson 2024, both used only 1ml ICG? sufficient or not enough?

Thank you very much for your comments. As mentioned in the previous response, these aspects are not entirely clarified and are based on the data from Xiong's study and the extrapolation of results from studies on endometrial cancer

5.  No image data were cited and compared in this systematic review, why?

Thank you for your question. The primary focus of this systematic review was to evaluate the various doses and volumes of indocyanine green (ICG) used in the detection of sentinel lymph nodes in cervical cancer. While image data can provide valuable insights, our review was centered specifically on summarizing the clinical outcomes related to the variation in doses and volumes. This approach allowed us to focus on these key factors, which are crucial for standardizing practice in this area. We recognize the importance of imaging data and may consider including this aspect in future research.

6.The statistical analysis was simple.

Thank you for your observation. The statistical analysis was intentionally kept simple to focus on providing a clear and straightforward comparison of the doses and volumes of indocyanine green (ICG) used across different studies. Our goal was to highlight the variations in clinical practice and offer insights that could be easily interpreted and applied. We believe this approach is appropriate for the scope of our review, but we are open to considering more complex statistical methods in future analyses if deemed necessary.

Reviewer 3 Report

Comments and Suggestions for Authors

Cervical cancer is one of the most common female cancers. It has been estimated that during one year 600000 new incidents will be noted. Fortunately, several years ago the vaccine was obtained moreover the diagnostic technique together with social awareness increases reduces the incidents. However, surgery is one of the main therapeutic treatments for cervical cancer. The main problem is the assignment/detection of metastasis to sentinel lymph nodes. The discussed in the article Indocyanine green was first used for endothelium cancer diagnosis and no strict recommendation for cervical cancer was made. Authors in their systematic review entitled: Impact of indocyanine green dose on sentinel lymph node mapping in cervical Cancer have been taken an effort to estimate under available data the concentration and dose volume of ICG. Their finding suggests that the 1.25 mg/ml in 2ml total volume of injection ICG give promising results. The proposed strategy increases the patient’s quality of Life after surgery which is important from a bioethics point of view. The references have been selected correctly. Moreover, the strategy of systematic review as well as the use of exclusion criteria and statistical analysis did not make my criticism and fears.

In conclusion, the article is a good starting point for further studies of ICG's broad application. Therefore. I recommend this review for publication in its current form.

Author Response

Thank you very much for your positive feedback and for recognizing the efforts made in our systematic review.

We are glad that our findings regarding the dose and volume of ICG are seen as promising and that you appreciate the impact this could have on improving the quality of life for cervical cancer patients.

We also appreciate your acknowledgment of the methodological approach we took in the review, including our selection of references, use of exclusion criteria, and statistical analysis.

Round 2

Reviewer 2 Report

Comments and Suggestions for Authors

The manuscript entitled "Impact of indocyanine green dose on sentinel lymph node mapping in cervical cancer" was re-submitted to Cancers for considerations. Nearly all my questions were answered or responded. I have no further concerns.